# High Expression of NRF2 Is Associated with Increased Tumor-Infiltrating Lymphocytes and Cancer Immunity in ER-Positive/HER2-Negative Breast Cancer

**DOI:** 10.3390/cancers12123856

**Published:** 2020-12-21

**Authors:** Masanori Oshi, Fernando A. Angarita, Yoshihisa Tokumaru, Li Yan, Ryusei Matsuyama, Itaru Endo, Kazuaki Takabe

**Affiliations:** 1Department of Surgical Oncology, Roswell Park Comprehensive Cancer Center, Buffalo, NY 14263, USA; masa1101oshi@gmail.com (M.O.); Fernando.AngaritaCelis@RoswellPark.org (F.A.A.); Yoshihisa.Tokumaru@roswellpark.org (Y.T.); 2Department of Gastroenterological Surgery, Yokohama City University Graduate School of Medicine, Yokohama 236-0004, Japan; ryusei@yokohama-cu.ac.jp (R.M.); endoit@med.yokohama-cu.ac.jp (I.E.); 3Department of Surgical Oncology, Graduate School of Medicine, Gifu University, 1-1 Yanagido, Gifu 501-1194, Japan; 4Department of Biostatistics & Bioinformatics, Roswell Park Comprehensive Cancer Center, Buffalo, NY 14263, USA; li.yan@roswellpark.org; 5Department of Surgery, Jacobs School of Medicine and Biomedical Sciences, State University of New York, Buffalo, NY 14263, USA; 6Division of Digestive and General Surgery, Niigata University Graduate School of Medical and Dental Sciences, Niigata 951-8520, Japan; 7Department of Breast Surgery, Fukushima Medical University School of Medicine, Fukushima 960-1295, Japan; 8Department of Breast Surgery and Oncology, Tokyo Medical University, Tokyo 160-8402, Japan

**Keywords:** biomarker, breast cancer, gene expression, hormonal, metastasis, *NRF2*, survival, treatment response

## Abstract

**Simple Summary:**

The clinical relevance of Nuclear factor erythroid 2-Related Factor 2 (*NRF2*) in human breast cancer remains unclear. A total of 5443 breast cancer patients with transcriptomic profile were analyzed for the clinical relevance of *NRF2* expression, including cancer aggressiveness, immune cell infiltration, patient survival, and drug response. We found that tumors with high *NRF2* expression were associated with better survival in ER-positive/HER2-negative breast cancer. *NRF2* expression was equivalent in immune, stromal, and cancer cells in tumor microenvironment. We found that high *NRF2* expression was associated with enhanced tumor-infiltrating lymphocytes in ER-positive/HER2-negative breast cancer. *NRF2* expression significantly correlated with drug sensitivity in multiple ER-positive breast cancer cell lines, but not associated with pathological complete response after neoadjuvant chemotherapy in breast cancer patients regardless of subtypes.

**Abstract:**

Nuclear factor erythroid 2-related factor 2 (*NRF2*) is a key modifier in breast cancer. It is unclear whether NRF2 suppresses or promotes breast cancer progression. We studied the clinical relevance of *NRF2* expression by conducting in silico analyses in 5443 breast cancer patients from several large patient cohorts (METABRIC, GSE96058, GSE25066, GSE20194, and GSE75688). *NRF2* expression was significantly associated with better survival, low Nottingham pathological grade, and ER-positive/HER2-negative and triple negative breast cancer (TNBC). High *NRF2* ER-positive/HER2-negative breast cancer enriched inflammation- and immune-related gene sets by GSEA. *NRF2* expression was elevated in immune, stromal, and cancer cells. High *NRF2* tumors were associated with high infiltration of immune cells (CD8^+^, CD4^+^, and dendritic cells (DC)) and stromal cells (adipocyte, fibroblasts, and keratinocytes), and with low fraction of Th1 cells. *NRF2* expression significantly correlated with area under the curve (AUC) of several drug response in multiple ER-positive breast cancer cell lines, however, there was no significant association between *NRF2* and pathologic complete response (pCR) rate after neoadjuvant chemotherapy in human samples. Finally, high *NRF2* breast cancer was associated with high expression of immune checkpoint molecules. In conclusion, *NRF2* expression was associated with enhanced tumor-infiltrating lymphocytes in ER-positive/HER2-negative breast cancer.

## 1. Introduction

Breast cancer is the most common cancer among women worldwide. Estrogen receptor (ER)-positive breast cancer is both the most common (>70%) and least aggressive subtype of breast cancer [1]. Unfortunately, the main challenges with this subtype include late recurrence, which occurs in 40% of patients more than 10 years after diagnosis [2]. These tumors also have a poor response to neoadjuvant chemotherapy (NAC) [3,4]. Therefore, it is necessary to develop a prognostic biomarker for ER-positive breast cancer to anticipate who will recur.

In recent years, several publications have described the role of nuclear factor erythroid 2-related factor 2 (*NRF2*) in cancer progression. Data lack on whether *NRF2* suppresses or promotes tumor progression [5,6,7]. Some of these findings lead to the question of whether *NRF2* should be targeted as an anti-cancer therapy [8]. *NRF2* is commonly known as a tumor suppressor because it protects cells from oxidative or electrophilic insults and is thus anti-carcinogenic. However, *NRF2* promotes survival of both normal and malignant cells. *NRF2* in normal cells activates broad defense mechanisms, such as elimination of reactive oxygen species (ROS), dampening of inflammation, drug and carcinogen detoxication, and intermediary metabolism [9,10], all of which protect cells from various carcinogenic stressors. There is cross-talk between *NRF2* signaling and NF-kB, p53, and Notch1 signaling, which affects cell survival [11]. On the other hand, constitutively elevated *NRF2* levels in cancer cells can enhance growth and develop chemotherapy resistance by creating a redox environment [12,13]. To this end, high levels of *NRF2* are generally correlated with poor prognosis in multiple types of cancer [5,6].

Limited data exist on *NRF2* activation in breast cancer. *NRF2* promotes breast cancer progression by enhancing glycolysis through coactivation of *HIF1α*, which allows some to suggest *NRF2* as a therapeutic target for breast cancer [14]. On the contrary, high *NRF2* gene expression was reported to be associated with better outcomes in ER-positive breast cancer [15]. Given that estrogen is known as a ROS inducer, ER-positive tumors may upregulate *NRF2* activity and accelerate their antioxidant response to resist oxidative stress [16]. Another study described a correlation between *NRF2* activity and survival outcome in ER-positive breast cancer compared to triple negative breast cancer (TNBC) [17]. To date, there is no study that has elucidated the mechanism in which *NRF2* activity in a bulk tumor is associated with better survival in ER-positive breast cancer patients.

Rapid evolution in microarray and sequencing technologies has revolutionized the depth and complexity at which molecular data are obtained and examined today in biomedical research. Although it is difficult to reproduce the human tumor microenvironment in vivo and vitro, it is possible to grasp several immune functions by using the gene expression of bulk tumor and algorithms. We previously reported on the relationship between transcriptome and clinical relevance with immunity and hallmark pathway by computational analysis using large patient cohorts using several algorithms, such as xCell and gene set variant analysis (GSVA) [18,19,20,21,22]. For example, we found the significant contribution of immune cells in a favorable survival outcome of high glucocorticoid receptor (GR) expression in ER-positive breast cancer [23]. We also established a 4-gene score based on genes differentially expressed between the parental and lung metastasis cell lines that predicts neoadjuvant chemotherapy (NAC) response in ER-positive/human epidermal growth factor receptor 2 (HER2)-negative breast cancer [24]. To this end, we are equipped with the necessary methods to analyze the clinical relevance of *NRF2* activity and its associated features in the tumor microenvironment.

We hypothesized that both cancer cells and immune cells in the tumor microenvironment contribute to the expression of *NRF2* in a bulk tumor. We also hypothesized that infiltration of immune cells may contribute to survival outcomes in breast cancer patients. The novelty of the study is to investigate the clinical relevance of *NRF2* using a large amount of human sample data.

## 2. Results

### 2.1. High NRF2 Expression Is Significantly Associated with Better Survival in ER-Positive/HER2-Negative Breast Cancer

Advances in technology and banking of transcriptomic data have been extremely rapid and robust in recent years. The large breast cancer patient cohort, METABRIC, was updated with clinical parameters, including longer survival follow-up. Another robust patient cohort (GSE96058) became available, which allows survival analyses by subtypes with stronger power. Thus, it was of interest to investigate how *NRF2* gene expression impacts different breast cancer subtypes using the latest METABRIC and GSE96058 cohorts. The top tertile was used as a cut-off between high and low *NRF2* groups within each cohort. High *NRF2* expression was significantly associated with better disease-free survival (DFS), disease-specific survival (DSS), and overall survival (OS) in ER-positive/HER2-negative breast cancer in the METABRIC cohort (Figure 1; *p* = 0.039, *p* = 0.011, and *p* < 0.001, respectively). The OS results were validated in the GSE96058 cohort (*p* = 0.018). On the other hand, although high *NRF2* expression in TNBC tends to be associated with better survival, there are no significant differences between high and low *NRF2* group in TNBC nor HER2-positive breast cancer in both patient cohorts.

### 2.2. High NRF2 Expression Is Significantly Associated with Lower Nottingham Pathological Grade, and ER-Positive/HER2-Negative and TNBC Subtype

Given the results of Figure 1, we anticipated that high *NRF2* expression would be associated with less aggressive clinical features. There was no trend of *NRF2* expression in the American Joint Commission on Cancer (AJCC) cancer staging. On the other hand, we found that higher *NRF2* expression was associated with lower Nottingham pathological grade, which reflects less cancer cell proliferation. High *NRF2* expression was associated with ER-positive/HER2-negative and TNBC subtypes. These results were consistent in both the METABRIC and GSE96058 cohorts (Figure 2).

### 2.3. High NRF2 Tumors Are Enriched with Inflammation- and Immune-Related Gene Sets in ER-Positive/HER2-Negative Breast Cancer

Given that high *NRF2* expression was associated with less cancer cell proliferation and with better survival outcomes in ER-positive/HER2-negative breast cancer, it was of interest to investigate the biological basis of this association in this subtype. Gene set enrichment analysis (GSEA) was used to study the enrichment of MSigDB Hallmark gene sets in breast cancers with high *NRF2*. We found that inflammation- and immune-related gene sets were significantly enriched in high *NRF2* tumors consistently in both the METABRIC and GSE96058 cohorts, such as inflammatory response, IL6/JAK/STAT3 signaling, tumor necrosis factor (TNF)-α signaling, complement, coagulation, allograft rejection, IL2/STAT5 signaling, interferon (IFN)-γ response, and apoptosis (Figure 3). Tumors with high *NRF2* expression also significantly expressed pro-cancerous gene sets, such as KRAS signaling up, TGF-β signaling, hypoxia, and angiogenesis, in both patient cohorts. Furthermore, tumors with high *NRF2* expression was significantly associated with high expression of inflammatory-related genes; *IL6, Lp-PLA2 (PLA2G7),* and Myeloperoxidase (*MPO)*, and apoptosis-related genes; *FAS, TNF, and TNFR1,* as well as high score of inflammatory-related gene sets; IL6/JAC/STAT3 signaling, and inflammatory response pathway, and apoptosis-related gene sets; Apoptosis and TNFα signaling Via NFkB pathway, which calculated by GSVA algorithm in both METABRIC and GSE96058 cohorts except for *TNF* and *TNFR1* expression in the METABRIC cohort (Appendix A). The METABRIC cohort did not have *MPO* expression data.

### 2.4. NRF2 Is Expressed in Immune Cells as well as Cancer Cells, and High NRF2 Tumors Are Infiltrated with Anti-Cancer Immune Cells and Stromal Cells

Given that tumors with high *NRF2* expression were associated with better survival outcomes and enriched inflammation- and immune-related gene sets in ER-positive/HER2-negative subtype, we hypothesized that non-cancer cells in the tumor microenvironment may be involved in *NRF2* expression of a bulk tumor. Single-cell sequencing technology allows transcriptomic profile and a better understanding of the function of an individual cell in the tumor microenvironment. Hence, a single-cell sequencing cohort of primary breast cancer (GSE75688) was used to analyze *NRF2* expression differences between immune (T cells, B cells, and myeloid cells), tumor, and stromal cells. Immune cells expressed NRF2 in equivalent levels as tumor cells (Figure 4A). In immune cells, *NRF2* expression levels were lower in B cells than in T cells or myeloid cells.

Therefore, it was of interest to investigate which immune cell in the tumor microenvironment was associated with *NRF2* gene expression in the bulk tumor. Immune and stromal cell compositions in bulk tumors were estimated using the xCell algorithm and compared between *NRF2* low and high groups in ER-positive/HER2-negative subtype of the METABRIC and GSE96058 cohorts. We found that *NRF2* high tumors had a significantly high fraction of CD8, CD4, and dendritic cells (DC), and low fraction of T helper 1 type cells (Th1) consistently in both cohorts (Figure 4B). M1 macrophages, natural killer, regulatory T cells, and T helper 2 type cells (Th2) were highly infiltrated in high *NRF2* tumors in the METABRIC cohort but not validated by the GSE96058 cohort.

Cytolytic activity score (CYT), defined as the sum of expression of granzyme A (*GZMA*) and perforin (PRF1), was used to evaluate overall anti-cancer immune cell killing in the tumor microenvironment (25594174). We found that *NRF2* high group was significantly associated with a high level of CYT consistently in both patient cohorts (Figure 4C, *p* < 0.001, and *p* = 0.004, respectively).

The high *NRF2* group was also significantly associated with a high fraction of stromal cells, including adipocytes, fibroblasts, and keratinocytes, consistently in both patient cohorts (Figure 4D). These results suggested that high *NRF2* expression was associated with a high fraction of anti-cancer immune cells with cytolytic activity as well as infiltration of stromal cells.

### 2.5. NRF2 Expression Was Significantly Associated with Treatment Response In Vitro, but Association Was Noted with Pathological Complete Response (pCR) after Neoadjuvant Chemotherapy (NAC)

Next, we investigated the relationship between *NRF2* expression and treatment response using in vitro data obtained from DepMap portal and patient cohorts that underwent NAC (GSE25066 and GSE20194). The levels of *NRF2* expression correlated with levels of area of under the curve (AUC) for paclitaxel, 5-FU (fluorouracil), tamoxifen, and fulvestrant in ER-positive/HER2-negative breast cancer cell lines (CAMA1, EFM19, HCC1428, HCC1500, KPL1, MCF7, MDAMB175VII, MDAMB415, T47D, and ZR751) (Figure 5A, *r* = 0.88 [*p* = 0.02], *r* = 0.86 [*p* < 0.01], *r* = 0.75 [*p* < 0.05], and *r* = 0.96 [*p* < 0.01], respectively). There were no significant differences between low and high *NRF2* groups in pCR rate after NAC in the GSE25066 and GSE20194 cohorts (Figure 5B). These results suggest that in vitro results of NRF2 expression may not be directly translatable to the clinical setting.

### 2.6. High NRF2 Tumors Are Significantly Associated with High Expression of Immune Checkpoint Molecules in ER-Positive/HER2-Negative Breast Cancer

Immunotherapy using immune checkpoint inhibitors is drawing attention as a new modality to treat cancer, however, it is only approved for TNBC. Therefore, it is necessary to develop a biomarker to identify which patients have a high expression of immune checkpoint molecules. We found that *NRF2* high tumors were significantly associated with higher expression of major immune checkpoint molecules, namely programmed death-1 (*PD-1*), programmed death ligand 2 (*PD-L2*), indoleamine dioxygenase 1 (*IDO1*), and B- and T-lymphocyte attenuator (*BTLA*). These findings were consistently noted in both patient cohorts (Figure 6). *NRF2* high tumors were also significantly associated with high expression of programmed death ligand 1 (*PD-L1*) and cytotoxic T-lymphocyte-associated protein 4 (*CTLA4*) in GSE96058 cohort.

## 3. Discussion

In this study, we found that high *NRF2* expression was significantly associated with better DFS, DSS, and OS in ER-positive/HER2-negative, but not in other subtypes across two large breast cancer patient cohorts (METABRIC and GSE96058). High *NRF2* expression was associated with low Nottingham pathological grade as well as ER-positive/HER2-negative and TNBC subtypes, but not with AJCC cancer staging. We found that high *NRF2* expression ER-positive/HER2-negative breast cancer significantly enriched inflammation- and immune-related gene sets as well as pro-cancerous gene sets by GSEA. Interestingly, NRF2 expression was elevated not only in cancer cells but also in T cells, myeloid cells, and stromal cells. High *NRF2* expression ER-positive/HER2-negative breast cancer was associated with increased tumor-infiltrating lymphocytes (CD8^+^ T cell, CD4^+^ T cell, and DC) and low fraction of Th1 cells. Several stromal cells, including adipocyte, fibroblasts, and keratinocytes, highly infiltrated tumors with high *NRF2* expression levels. CYT, which assesses overall immune cytolytic activity, was elevated in high *NRF2* expression tumors. *NRF2* expression levels correlated with AUC of several drug response in vitro, however, there was no association between *NRF2* expression and pCR rate after NAC in two cohorts. Finally, high *NRF2* expression ER-positive/HER2-negative breast cancer was associated with high expression of immune checkpoint molecules.

The novelty of the study is to investigate the clinical relevance of *NRF2* using a large amount of human sample data. Recent advances in high-throughput technology resulted in accumulation of tumor transcriptome data of large sample size cohorts. There has been a dramatic advancement in the use of genetic analysis for cancer research due to the availability of data collected from around the world, including projects such as METABRIC and data repositories such as Gene Expression Omnibus (GEO). The clinical outcomes of these databases are updated and allow researchers to analyze long-term follow-up data. For example, the currently available median clinical follow-up is 14 years [25]. This is particularly relevant for ER-positive breast cancer that often relapses more than a decade after diagnosis. Recent advances in gene analysis technology allow us to obtain robust information from transcriptome data in bulk tumor by using algorithms which are released on a monthly basis. The tumor immune microenvironment, which plays a significant role in cancer progression and treatment response, has been analyzed traditionally by flow cytometry or immunohistochemistry. These approaches are highly labor-intensive and expensive when they are used to analyze thousands of patient samples, whereas computational algorithms can estimate the quantity of immune cells from tens of thousands of samples with less cost and much quicker [26,27,28,29] (as long as transcriptomic data are available). Furthermore, analyses such as GSEA explore the biological activity of a signaling pathway of interest and allow investigators to grasp the big picture of the intricately intertwined gene signaling pathways and to identify the mechanism that is in place. The clinical relevance of *NRF2* expression in tumors remains controversial in breast cancer. Wofl et al. suggested that *NRF2* expression of a bulk tumor may be useful as a predictive biomarker for ER-positive breast cancer using the METABRIC cohort in 2016 [15]. In this study, we have used the updated METABRIC cohort and obtained a similar result that patients with high *NRF2* expression ER-positive breast tumor have a better prognosis. Importantly, this result was validated in a larger sample size cohort (GSE96058). Somewhat surprisingly, we found that not only cancer cells but also immune cells and stromal cells were expressing high levels of *NRF2* gene in tumor microenvironment. NRF2 activation has also been reported to play a critical role in proper immune function. *NRF2* suppresses macrophage inflammatory response by blocking the transcription of proinflammatory cytokines, including IL-6 and IL-1b. This was found to be independent of *ROS* and the canonical *NRF2*-binding motif in cytokine genes [30]. The regulation of the immune microenvironment also extends beyond macrophages to other myeloid populations and regulatory T cells [31,32], which can influence tumor progression and metastasis. *NRF2* activation within the tumor microenvironment suppressed the progression of lung tumors [33]. Hayashi et al. reported that microenvironmental *NRF2* activation suppresses the progression of malignant *NRF2*-acitvated tumors and that *NRF2* activation in immune cells at least partially contributes to these suppressive effects [33]. Rong et al. reported that sulforaphane blocked prostaglandin E2 synthesis in parental and doxorubicin-resistant breast cancer 4T1 cell lines by activating *NRF2*, and triggered MDSCs to switch to an immunogenic phenotype, enhancing the anti-tumor activities of CD8^+^ T cells [34]. High immune activity is known to be associated with a better prognosis in many cancers, including breast cancer [35,36,37]. In vivo/vitro preclinical models are essential tools to elucidate cancer biology. With that said, we are also aware that no model can perfectly replicate human cancer. Therefore, in this study, we investigated the association between *NRF2* expression and immunity in tumor microenvironment using large amounts of human sample data to see if what has been reported actually occurs in human tumors. Considering that high *NRF2* expression ER-positive breast tumor was significantly associated with increased tumor-infiltrating lymphocytes, especially anti-cancer immune cells, CD8^+^ T cell, CD4^+^ T cell, and DC cells, we speculate that the better survival of high *NRF2* expression ER-positive/HER2-negative breast cancer may be at least partly due to the reflection of high infiltration of these immune cells. This notion is in agreement with our other results that high *NRF2* expression tumor enriched immune-related gene sets and was associated with overall cytolytic activity.

*NRF2* is generally known as a tumor suppressor and several *NRF2* activators are currently being tested as chemopreventive compounds in clinical trials. On the other hand, there are serious clinical concerns on enhanced *NRF2* activity because it may also protect cancer cells from chemotherapeutic agents and facilitate cancer progression as *NRF2* protects normal cells. These studies have included a diverse range of drugs, such as cisplatin, carboplatin, 5-fluorouracil, paclitaxel, bleomycin, doxorubicin, and etoposide [38,39,40]. Thus, the role of *NRF2* in cancer is subject of controversial discussion, as it has been reported to have both pro- and anti-tumorigenic functions [15]. Furthermore, several papers reported that the role of *NRF2* may be context-dependent, because of complex *NRF2*-related pathways [41,42]. One of the reasons for this is that the tumor environment is not constant in the in vivo and in vitro settings. Moreover, it is also difficult to reproduce the human tumor environment in vivo and in vitro. Several efforts have been made or are ongoing to develop novel therapeutics, but they have been hampered by the lack of preclinical models that reliably reproduce the human tumor microenvironment. We previously reported the utility of patient-derived xenograft mouse model using human breast cancer patient samples, however, it was not possible to replicate human tumors. Our results, which showed the different result of the association of *NRF2* expression with treatment response between in vitro and neoadjuvant human patient data, indicate that the role of non-tumor cells in a bulk tumor, such as immune cells, may have clinical relevance. Additionally, this further emphasizes the importance of analyzing clinical specimens. We believe that our in silico approach is a useful tool to obtain a comprehensive view of human cancers in the clinical setting.

Although immune checkpoint inhibitors are approved for advanced breast cancer, their indication is very limited, and patient selection remains a major challenge. Given that our results show that the majority of immune checkpoint molecules were associated with high *NRF2* expression in ER-positive/HER2-negative breast cancer, we cannot help but speculate that patients whose tumors show high *NRF2* expression could be the population who may respond to immune checkpoint inhibitors even though they may not respond to chemotherapy.

Although we found clinical relevance of *NRF2* expression in breast cancer using several algorithms, this study is not free from limitations. The biggest limitation is that our analysis is a retrospective study and limited in the measurement of gene expressions alone. Experimental approaches are needed to elucidate the mechanism in the future. Especially the relationship between the *NRF2* expression and immunity should be directly quantified in tumoral immune cells using a gold standard such as flow cytometry or immunohistochemistry. Furthermore, to use *NRF2* expression as a predictive biomarker in ER-positive/HER2-negative management, we need to conduct a prospective clinical trial.

In conclusion, we demonstrated that immune cells in addition to tumor cells express high levels of *NRF2*, and high *NRF2* expression enriched inflammation- and immune-related gene sets and was associated with enhanced tumor-infiltrating lymphocytes in ER-positive/HER2-negative breast cancer, which may at least partly explain why high *NRF2* expression was associated with better survival in that subtype.

## 4. Materials and Methods

### 4.1. Cohorts Used for Analyses

For the main analysis, we used the Molecular Taxonomy of Breast Cancer International Consortium (METABRIC) (*n* = 1903) [25,43], and the GSE96058 (*n* = 3234) [44] cohorts. Both these cohorts have a large number of breast cancer samples with tumor transcriptome and clinical data. The METABRIC cohort data were obtained from the cBio Cancer Genomic portal [45]. The GSE96058 cohort data were obtained from the Gene Expression Omnibus (GEO) repository. We also obtained other GEO data from the GSE25066 (*n* = 508) [46] andGSE20194 (*n* = 248) [47] cohorts. Single cell sequencing data were obtained from primary breast cancer tumor from the GSE75688 cohort [48].

### 4.2. Data of Gene Expression and Treatment Response of Cell Lines

Gene expression and level of AUC of several drugs (paclitaxel, 5-FU, tamoxifen, and fulvestrant) of ER-positive/HER2-negative breast cancer cell lines were obtained through DepMap portal (https://depmap.org/portal/). Cell lines containing AUC data for each drug were used. ER-positive/HER2-negative breast cancer cell lines included CAMA1, EFM19, HCC1428, HCC1500, KPL1, MCF7, MDAMB175VII, MDAMB415, T47D, and ZR751.

### 4.3. Tumor-Infiltrating Cells Scoring Using xCell Algorithm and Cytolytic Activity Score (CYT)

The xCell, a bioinformatics algorithm [29], was used to predict immune composition in the METABRIC and GSE96058 cohort samples. A set of 64 cell reference profiles were used. A signature to predict their absolute levels within each sample was developed, as we described previously [49,50,51,52,53,54,55,56,57]. The cytolytic activity score (CYT) as defined by Rooney et al. was used in the algorithm using the gene expression levels of granzyme A (*GZMA*) and perforin (*PRF1*) published in Cell, 2015 [58].

### 4.4. Gene Set Expression Analyses

Gene set enrichment analyses were performed using Gene Set Enrichment Analyses (GSEA) software (Java version 4.0) [59] with MSigDB Hallmark gene sets [60]. Statistical significance was defined by a false discovery rate (FDR) of 0.25, as recommended by the GSEA software.

### 4.5. Other

The R software (version 4.0.1, R Project for Statistical Computing) was used for all analyses. The top tertile for tumor *NRF2* expression was divided into high and low *NRF2* groups within cohorts. For group comparison, a one-way analysis of variance (ANOVA) and Fisher’s exact test were used. The Kaplan–Meier method and log-rank test were used for survival analyses. Boxplots were used to depict median and inter-quartile level values.

## 5. Conclusions

We found that immune cells, in addition to tumor cells, express high levels of *NRF2* in the tumor microenvironment. High *NRF2* expression levels enriched inflammation- and immune-related gene sets and were associated with enhanced tumor-infiltrating lymphocytes in ER-positive/HER2-negative breast cancer. These results may explain why high *NRF2* expression was associated with better survival outcomes in this subtype.

## Figures and Tables

**Figure 1 cancers-12-03856-f001:**
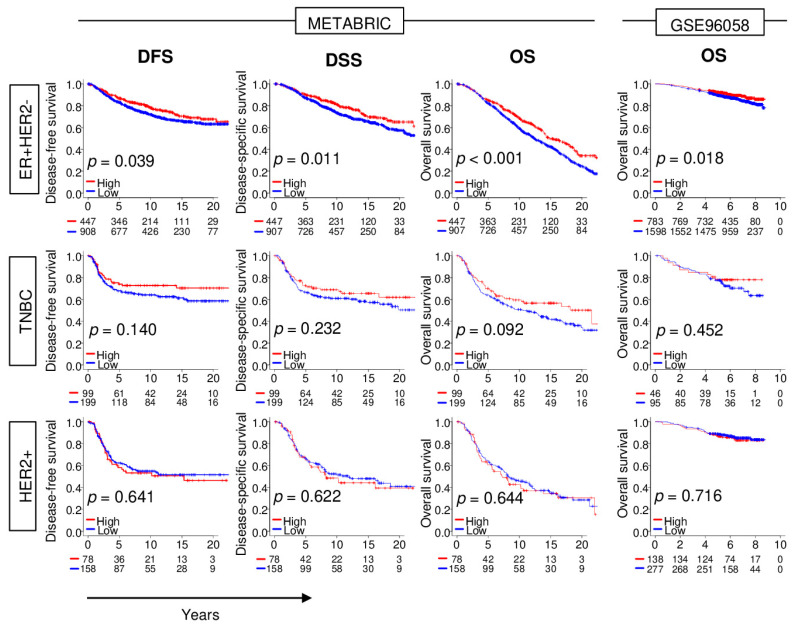
Association between levels of expression of *NRF2* and survival outcomes of breast cancer patients in the METABRIC and GSE96058 cohorts. Disease-free survival (DFS), disease-specific survival (DSS), and overall survival (OS) of *NRF2* low (blue) and high (red) of *NRF2* expression by breast cancer subtype.

**Figure 2 cancers-12-03856-f002:**
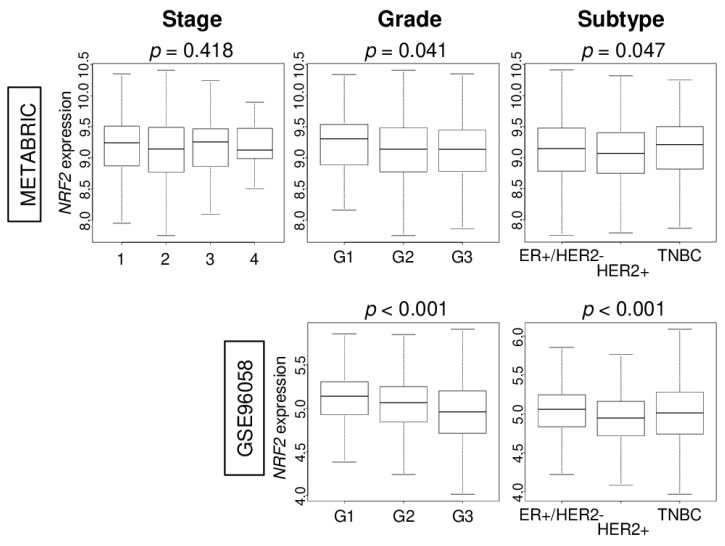
Association of *NRF2* levels of expression according to breast cancer clinical features in the METABRIC and GSE96058 cohorts. Boxplots of the *NRF2* expression by breast cancer American Joint Committee on Cancer (AJCC) stage, Nottingham pathological grade, and breast cancer subtypes [estrogen receptor-positive/human epidermal growth factor receptor 2-negative (ER+/HER2), triple negative breast cancer (TNBC), and HER2-positive].

**Figure 3 cancers-12-03856-f003:**
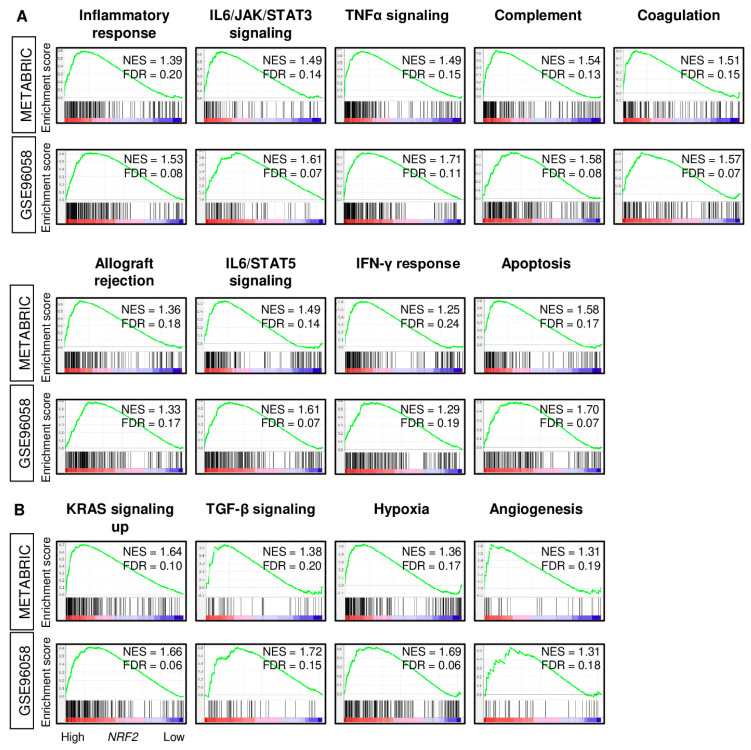
Gene Set Enrichment Assay (GSEA) with enrichment gene sets in the *NRF2* high expression group of ER-positive/HER2-negative patients from the METABRIC and GSE96058 cohorts. (**A**) Inflammation- and immune-related gene sets (inflammatory response, IL6/JAK/STAT3 signaling, TNF-α signaling, complement, coagulation, allograft rejection, IL2/STAT5 signaling, interferon (IFN)-γ response, and apoptosis and (**B**) Pro-cancerous-related gene sets (KRAS signaling up, TGF-β signaling, hypoxia, and angiogenesis) with normalized enrichment score (NES) and false discovery rate (FDR). Statistical significance was defined by an FDR of 0.25, as recommended by the GSEA software.

**Figure 4 cancers-12-03856-f004:**
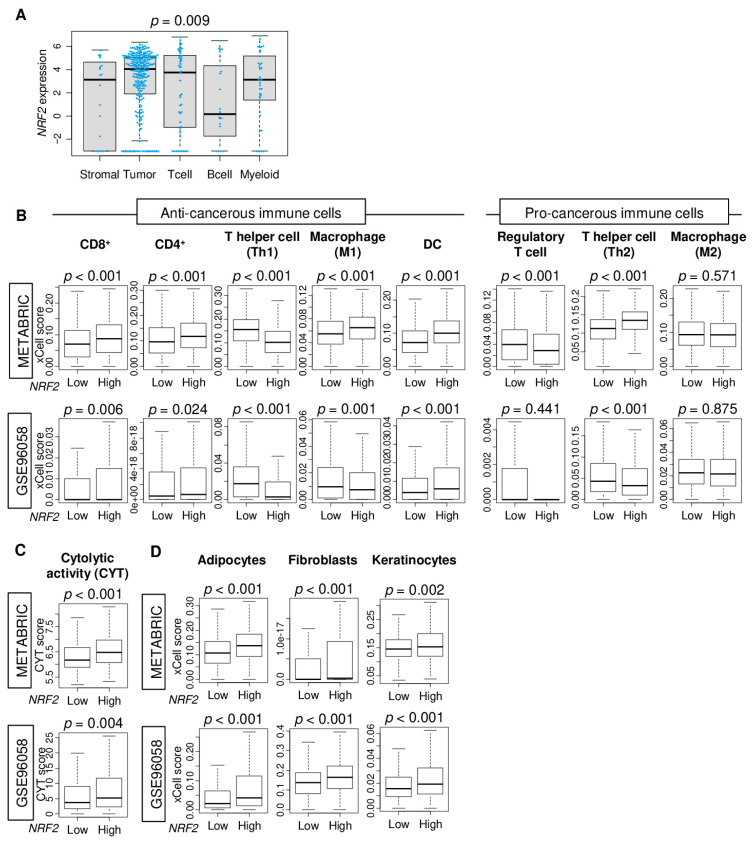
Association of *NRF2* expression level with fraction of immune cells and cytolytic activity (CYT). (**A**) Boxplots comparing *NRF2* expression levels by tumor cells, T cells, B cells, myeloid cells, and stromal cells in the GSE75688 cohort. Boxplots depicting the fraction of (**B**) immune cells, (**C**) CYT, and (**D**) stromal cells by *NRF2* low and high groups in METABRIC and GSE96058 cohorts. The cut-off of top tertile of *NRF2* expression was considered as *NRF2* high and low within each cohort. One-way ANOVA test was used for comparison.

**Figure 5 cancers-12-03856-f005:**
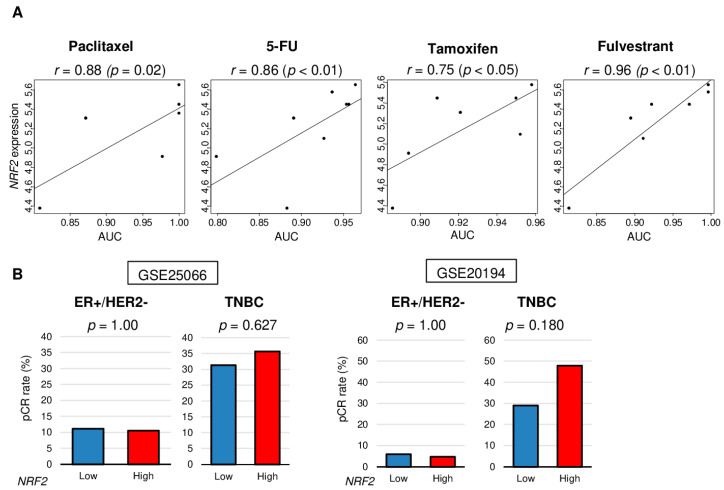
Association of *NRF2* expression level with treatment response of cell lines and human tumors. (**A**) Correlation plots between *NRF2* expression level and area under the curve (AUC) of several drug sensitivity for ER-positive/HER2-negative breast cancer cell lines. Spearman correlation statistics are depicted. (**B**) Bar plots comparing the pathologic complete response (pCR) rates between low and high NRF2 groups among patients with ER-positive/HER2-negative tumors and triple negative breast cancer (TNBC) in the GSE25066 (*n* = 508), and GSE20194 (*n* = 278) cohorts. Fisher’s exact test was used to compare groups.

**Figure 6 cancers-12-03856-f006:**
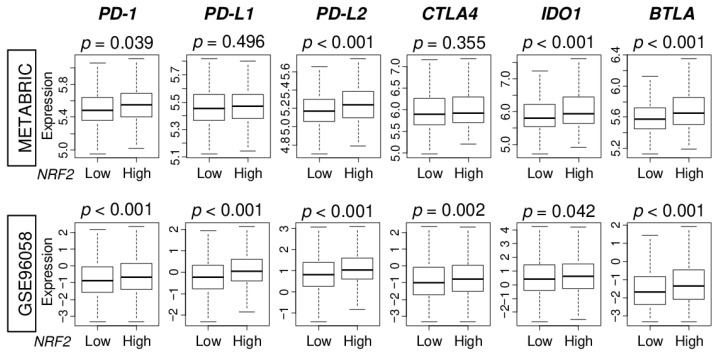
Association of NRF2 expression with immune checkpoint molecules in the METABRIC and GSE96058 cohorts. Boxplots comparing low and high NRF2 group with expression levels of immune checkpoint molecules [programmed death-1; PD-1, programmed death ligand 1; PD-L1, programmed death ligand 2; PD-L2, cytotoxic T-lymphocyte-associated protein 4; CTLA4, indoleamine dioxygenase 1; IDO1, B- and T-lymphocyte attenuator; BTLA] in ER-positive/HER2-negative breast cancer patients.

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
