# Peer review of "High Expression of NRF2 Is Associated with Increased Tumor-Infiltrating Lymphocytes and Cancer Immunity in ER-Positive/HER2-Negative Breast Cancer"

_cancers, 2020, doi:10.3390/cancers12123856_

Round 1

Reviewer 1 Report

The manuscript has been appropriately revised  to address concerns. I have no further comments at this time.

Reviewer 2 Report

Although the study still lacks novelty/originality, the revised manuscript better presents the details on enhanced NRF2 expression with immunity in ER-positive/HER2-negative breast cancer. I don't have any further comments on this MS.

Reviewer 3 Report

The authors modified the manuscript in relation to the requests.
In my opinion the new version of the paper is now acceptable for publication

in Cancers 

This manuscript is a resubmission of an earlier submission. The following is a list of the peer review reports and author responses from that submission.

Round 1

Reviewer 1 Report

The authors have analyzed publicly available genomic dataset to associate NRF2 expression with higher immune infiltration and possibly better survival in a select cohort of breast cancer patients. The study is mostly associative and lacks originality. Overexpression of NRF2 has been long associated with chemo resistance and radiotherapy resistance including poor survival in many ER positive patients. Several findings have been published pointing to its role in Triple negative breast cancer where again high expression of NRF2 has been associated with a poor response. In the light of previously published results, these new findings are provocative but cannot be definitively accepted without any experimental proof.

Reviewer 2 Report

This is an interesting study where the investigators sought for the expression of NRF2 in breast cancer patients and found that high NRF2 expression levels promoted tumor infiltrating lymphocytes in ER+/HER2- breast cancer. Overall, the study shows some promising results such as, high NRF2 expression correlates with better survival, enriched with inflammatory and immune gene expression, more anti-cancer immune cells etc. However, the following are my suggestions on the manuscript: 

  1. Please correct some minor grammatical/typo (e.g. Data lacks on whether NRF2 is suppresses or promotes tumor progression [5-7])
  2. There is not enough mechanistic explanation on why higher NRF2 expression might offer a greater survival in ER+/HER2- breast cancer subtype. This would be really helpful for the readers if you could please provide slightly more discussion on this point.

Reviewer 3 Report

The paper of Oshi et al. is a retrospective analysis  exclusively based on the evaluation of the expression of certain genes in different cohorts of breast cancer patients. The study shows that in ER positive /Her negative breast cancers there is a high expression of NRF2 that correlates with an increased survival. In addition, the authors point to a correlation between high levels of NRF2 and the expression of genes indexing an inflammatory response and the involvement of immune system cells. The study is interesting but, in my opinion, limited. There is no evidence that high levels of NRF2 are responsible for the increased immune and inflammatory response.  This aspect could be analysed in vitro by assessing in  breast cancer cells the levels of both NRF2 and some inflammatory markers, as well as their correlation using Nrf2 activators or inhibitors. Finally, the paper refers to an increase in apoptosis, but the apoptotic  markers analysed are not described